# Sample Complexity of Algorithm Selection Using Neural Networks and Its Applications to Branch-and-Cut

**Hongyu Cheng**
Dept. of Applied Mathematics & Statistics
Johns Hopkins University
Baltimore, MD 21218
hongyucheng@jhu.edu

**Sammy Khalife**
Dept. of Applied Mathematics & Statistics
Johns Hopkins University
Baltimore, MD 21218
khalife.sammy@jhu.edu

**Barbara Fiedorowicz**
Dept. of Applied Mathematics & Statistics
Johns Hopkins University
Baltimore, MD 21218
bfiedor1@jhu.edu

**Amitabh Basu**
Dept. of Applied Mathematics & Statistics
Johns Hopkins University
Baltimore, MD 21218
basu.amitabh@jhu.edu

## Abstract

Data-driven algorithm design is a paradigm that uses statistical and machine learning techniques to select from a class of algorithms for a computational problem an algorithm that has the best expected performance with respect to some (unknown) distribution on the instances of the problem. We build upon recent work in this line of research by considering the setup where, instead of selecting a single algorithm that has the best performance, we allow the possibility of selecting an algorithm based on the instance to be solved, using neural networks. In particular, given a representative sample of instances, we learn a neural network that maps an instance of the problem to the most appropriate algorithm *for that instance*. We formalize this idea and derive rigorous sample complexity bounds for this learning problem, in the spirit of recent work in data-driven algorithm design. We then apply this approach to the problem of making good decisions in the branch-and-cut framework for mixed-integer optimization (e.g., which cut to add?). In other words, the neural network will take as input a mixed-integer optimization instance and output a decision that will result in a small branch-and-cut tree for that instance. Our computational results provide evidence that our particular way of using neural networks for cut selection can make a significant impact in reducing branch-and-cut tree sizes, compared to previous data-driven approaches.

## 1 Background and motivation

Often there are several competing algorithms for a computational problem and no single algorithm dominates all the others. The choice of an algorithm in such cases is often dictated by the "typical" instance one expects to see, which may differ from one application context to another. Data-driven algorithm design has emerged in recent years as a way to formalize this question of algorithm selection and draws upon statistical and machine learning techniques; see Balcan [2020] for a survey and references therein. More formally, suppose one has a class $\mathcal{I}$ of instances of some computational problem with some unknown distribution, and a class of algorithms that can solve this problem parameterized by some "tunable parameters". Suppose that for each setting of the parameters, the corresponding algorithm can be evaluated by a score function that tells us how

38th Conference on Neural Information Processing Systems (NeurIPS 2024).

well the algorithm does on different instances (e.g., running time, memory usage etc.). We wish to find the set of parameters that minimizes (or maximizes, depending on the nature of the score) the expected score on the instances (with respect to the unknown distribution), after getting access to an i.i.d. sample of instances from the distribution. For example, $\mathcal{I}$ could be a family of mixed-integer optimization problems and the class of algorithms are branch-and-cut methods with their different possible parameter settings, and the score function could be the size of the branch-and-cut tree.

In Balcan et al. [2021a], the authors prove a central result on the sample complexity of this problem: if for any fixed instance in $\mathcal{I}$, the score function has a piecewise structure as a function of the parameters, where the number of pieces is upper bounded by a constant $R$ independent of the instance, the pieces come from a family of functions of "low complexity", and the regions partitioning the parameter space into pieces have "low complexity" in terms of their shape, then the sample complexity is a precise function of $R$ and these two complexity numbers (formalized by the notion of pseudo-dimension – see below). See Theorem 3.3 in Balcan et al. [2021a] for details. The authors then proceed to apply this key result to a diverse variety of algorithmic problems ranging from computational biology, computational economics to integer optimization.

Our work in this paper is motivated by the following consideration. In many of these applications, one would like to select not a *single* algorithm, i.e., a single setting of the parameters, but would like to decide on the parameters after receiving a new instance of the problem. In other words, we would like to learn not the best parameter, but the *best mapping from instances to parameters*. We consider this version of the problem, where the mapping is taken from a class of neural network functions. At a high level, switching the problem from finding the "best" parameter to finding the "best" mapping in a parameterized family of mappings gives the same problem: we now have a new set of parameters – the ones parameterizing the mappings (neural networks) – and we have to select the best parameter. And indeed, the result from Balcan et al. [2021a] can be applied to this new setting, *as long as one can prove good bounds on pseudo-dimension in the space of these new parameters*. The key point of our result is two fold:

1. Even if original space of parameters for the algorithms is amenable to the analysis of pseudo-dimension as done in Balcan et al. [2021a], it is not clear that this immediately translates to a similar analysis in the parameter space of the neural networks, because the "low complexity piecewise structure" from the original parameter space may not be preserved.

2. We suspect that our proof technique results in tighter sample complexity bounds, compared to what one would obtain if one could do the analysis from Balcan et al. [2021a] in the parameter space of the neural network directly. However, we do not have quantitative evidence of this yet because their analysis seems difficult to carry out directly in the neural parameter space.

One of the foundational works in the field of *selecting algorithms based on specific instances* is Rice [1976], and recently, Gupta and Roughgarden [2016], Balcan et al. [2021c] have explored the sample complexity of learning mappings from instances to algorithms for particular problems. Our approach is also related to recent work on *algorithm design with predictions*; see Mitzenmacher and Vassilvitskii [2022] for a short introduction and the references therein for more thorough surveys and recent work. However, our emphasis and the nature of our results are quite different from the focus of previous research. We establish sample complexity bounds for a general learning framework that employs neural networks to map instances to algorithms, which is suitable for handling some highly complex score functions, such as the size of the branch-and-cut tree. The introduction of neural networks in our approach bring more intricate technical challenges than traditional settings like linear predictors or regression trees Gupta and Roughgarden [2016]. On the application side, as long as we know some information about the algorithms (see Theorems 2.5 and 2.6), our results can be directly applied. This is demonstrated in this paper through applications to various *cutting plane selection* problems in branch-and-cut in Propositions 3.3, 3.5, 3.6 and 3.7. Several references to fundamental applied work in data-driven algorithm selection can be found in the Balcan [2020], Gupta and Roughgarden [2016], Balcan et al. [2021a,c].

## 1.1 Applications in branch-and-cut methods for mixed-integer linear optimization

Mixed-integer linear optimization is a powerful tool that is used in a diverse number of application domains. Branch-and-cut is the solution methodology of choice for all state-of-the-art solvers for

mixed-integer optimization that is founded upon a well-developed theory of convexification and systematic enumeration Schrijver [1986], Nemhauser and Wolsey [1988], Conforti et al. [2014]. However, even after decades of theoretical and computational advances, several key aspects of branch-and-cut are not well understood. During the execution of the branch-and-cut algorithm on an instance, the algorithm has to repeatedly make certain decisions such as which node of the search tree to process next, whether one should branch or add cutting planes, which cutting planes to add, or which branching strategy to use. The choices that give a small tree size for a particular instance may not be good choices for a different instance and result in a much larger tree. Thus, adapting these choices to the particular instance can be beneficial for overall efficiency. Of course, the strategies already in place in the best implementations of branch-and-cut have to adapt to the instances. For example, certain cutting plane choices may not be possible for certain instances. But even beyond that, there are certain heuristics in place that adapt these choices to the instance. These heuristics have been arrived at by decades of computational experience from practitioners. The goal in recent years is to provide a *data driven* approach to making these choices. In a recent series of papers Balcan et al. [2021a,d,b, 2022, 2018], the authors apply the general sample complexity result from Balcan et al. [2021a] in the specific context of branch-and-cut methods for mixed-integer linear optimization to obtain several remarkable and first-of-their-kind results. We summarize here those results that are most relevant to the cut selection problem since this is the focus of our work.

1. In Balcan et al. [2021d], the authors consider the problem of selecting the best Chvátal-Gomory (CG) cutting plane (or collection of cutting planes) to be added at the root node of the branch-and-cut tree. Thus, the "tunable parameters" are the possible multipliers to be used to generate the CG cut at the root node. The score function is the size of the resulting branch-and-cut tree. The authors observe that for a fixed instance of a mixed-integer linear optimization problem, there are only finitely many CG cuts possible and the number can be bounded explicitly in terms of entries of the linear constraints of the problem. Via a sophisticated piece of analysis, this gives the required piecewise structure on the space of multipliers to apply the general result explained above. This gives concrete sample complexity bounds for choosing the multipliers with the best expected performance. See Theorem 3.3, 3.5 and 3.6 in Balcan et al. [2021d] for details. Note that this result is about selecting a *single* set of multipliers/cutting planes that has the best expected performance across all instances. This contrasts with selecting a good strategy to select multipliers *depending on the instance*, that has good expected performance (see point 2. below).

2. The authors in Balcan et al. [2021d] also consider the problem of learning a good strategy that selects multipliers based on the instance. In particular, they consider various auxiliary score functions used in integer optimization practice, that map a pair of instance $I$ and a cutting plane $c$ for $I$ that measures how well $c$ will perform for processing $I$. The strategy will be a linear function of these auxiliary scores, i.e., a weighted average of these auxiliary scores, and the learning problem becomes the problem of finding the best linear coefficients for the different auxiliary scores. So now these linear coefficients become the "tunable parameters" for the general result from Balcan et al. [2021a]. It is not hard to find the piecewise structure in the space of these new parameters, given the analysis in the space of CG cut multipliers from point 1. above. This then gives concrete sample complexity bounds for learning the best set of weights for these auxiliary score functions for cut selection. See Theorem 4.1 and Corollary 4.2 in Balcan et al. [2021d] for details.

3. In all the previous results discussed above, the cutting planes considered were CG cuts, or it was essentially assumed that there are only a finite number of cutting planes available at any stage of the branch-and-cut algorithm. In the most recent paper Balcan et al. [2022], the authors consider general cutting plane paradigms, and also consider the possibility of allowing more general strategies to select cutting planes beyond using weighted combinations of auxiliary score functions. They uncover the subtlety that *allowing general mappings from instances to cutting planes can lead to infinite sample complexity and learning such mappings could be impossible, if the class of mappings is allowed to be too rich*. See Theorem 5.1 in Balcan et al. [2022]. This point will be important when we discuss our approach below.

   On the positive side, they show that the well-known class of Gomory-Mixed-Integer (GMI) cuts has a similar structure to CG cuts, and therefore, using similar techniques as discussed above, they derive sample complexity bounds for selecting GMI cuts at the root node. See Theorem 5.5 in Balcan et al. [2022]. As far as we understand, the analysis should extend to

the problem of learning weighted combinations of auxiliary score functions to select the GMI cuts as well using the same techniques as Balcan et al. [2021d], although the authors do not explicitly do this in Balcan et al. [2022].

**Our approach and results.**   Our point of departure from the line of work discussed above is that instead of using weighted combinations of auxiliary scores to select cutting planes, we wish to select these cutting planes using neural networks that map instances to cutting planes. In other words, in the general framework described above, the "tunable parameters" are the weights of the neural network. The overall score function is the size of the branch-and-cut tree after cuts are added at the root. We highlight the two main differences caused by this change in perspective.

1. In the approach where weighted combinations of auxiliary score functions are used, after the weights are learnt from the sample instances, for every new/unseen instance one has to compute the cut that maximizes the weighted score. This could be an expensive optimization problem in its own right. In contrast, with our neural approach, after training the net (i.e., learning the weights of the neural network), any new instance is just fed into the neural network and the output is the cutting plane(s) to be used for this instance. This is, in principle, a much simpler computational problem than optimizing the combined auxiliary score functions over the space of cuts.

2. Since we use the neural network to directly search for a good cut, bypassing the weighted combinations of auxiliary scores, we actually are able to find better cuts that reduce the tree sizes by a significant factor, compared to the approach of using weighted combinations auxiliary scores.

The above two points are clearly evidenced by our computational investigations which we present in Section A. The theoretical sample complexity bounds for cut selection are presented in Section 3. As mentioned before, these are obtained using the main sample complexity result for using neural networks for data driven algorithm design that is presented in Section 2.

**Comparison with prior work on cut selection using learning techniques.**   As already discussed above, our theoretical sample complexity work in cut selection is closest in spirit to the work in Balcan et al. [2021a,d,b, 2022, 2018]. However, there are several other works in the literature that use machine learning ideas to approach the problem of cut selection; see Deza and Khalil [2023] for a survey. Tang et al. Tang et al. [2020] initiated the exploration of applying Reinforcement Learning (RL) to select CG cuts derived from the simplex tableau. Huang et al. Huang et al. [2022] apply supervised learning to rank a "bag of cuts" from a set of cuts to reduce the total runtime. More recently, the authors in Turner et al. [2023] propose a novel strategy that use RL and Graph Neural Networks (GNN) to select instance-dependent weights for the combination of auxiliary score functions.

Our computational investigations, as detailed in Appendix A, distinguishes itself from these prior computational explorations in several key aspects:

1. Previous methods were limited to a finite set of candidate cuts, requiring either an optimal simplex tableau or relying on a finite collection of combinatorial cuts. In contrast, our approach allows the possibility of selecting from a possibly infinite family of cutting planes. Moreover, our method eliminates the need for computing a simplex tableau which can lead to a significant savings in computation (see Table 1 and the discussion in Section A.2).

2. Many prior studies aimed at improving the objective value rather than directly reducing the branch-and-cut runtime—with the exception of Huang et al. Huang et al. [2022], who explored this through supervised learning. To the best of our knowledge, our RL-based model is the first to directly target the reduction of the branch-and-cut tree size as its reward metric, which is strongly correlated with the overall running time.

3. Prior deep learning approaches are not underpinned by theoretical guarantees, such as sample complexity bounds. Our empirical work takes the theoretical insights for the branch-and-cut problem presented in Theorem 2.3 and Proposition 3.3 as its basis.

The limitations of our approach are discussed in Section 4.

# 2 Formal statement of results

We denote $[d]$ as the set $\{1, 2, \ldots, d\}$ for any positive integer $d \in \mathbb{Z}_+$. For a set of vectors $\{\mathbf{x}^1, \ldots, \mathbf{x}^t\} \subseteq \mathbb{R}^d$, we use superscripts to denote vector indices, while subscripts specify the coordinates in a vector; thus, $\mathbf{x}_j^i$ refers to the $j$-th coordinate of $\mathbf{x}^i$. Additionally, the sign function, denoted as $\mathrm{sgn} : \mathbb{R} \to \{0, 1\}$, is defined such that for any $x \in \mathbb{R}$, $\mathrm{sgn}(x) = 0$ if $x < 0$, and 1 otherwise. This function is applied to each entry individually when applied to a vector. Lastly, the notation $\lfloor \cdot \rfloor$ is used to indicate the elementwise floor function, rounding down each component of a vector to the nearest integer.

## 2.1 Preliminaries

### 2.1.1 Background from learning theory

**Definition 2.1** (Parameterized function classes). A *parameterized function class* is given by a function defined as
$$h : \mathcal{I} \times \mathcal{P} \to \mathcal{O},$$
where $\mathcal{I}$ represents the *input space*, $\mathcal{P}$ denotes the *parameter space*, and $\mathcal{O}$ denotes the *output space*. For any fixed parameter setting $\mathbf{p} \in \mathcal{P}$, we define a function $h_{\mathbf{p}} : \mathcal{I} \to \mathcal{O}$ as $h_{\mathbf{p}}(I) = h(I, \mathbf{p})$ for all $I \in \mathcal{I}$. The set of all such functions defines the parameterized function class, a.k.a. the *hypothesis class* $\mathcal{H} = \{h_{\mathbf{p}} : \mathcal{I} \to \mathcal{O} \mid \mathbf{p} \in \mathcal{P}\}$ *defined by* $h$.

**Definition 2.2** (Pseudo-dimension). Let $\mathcal{F}$ be a non-empty collection of functions from an input space $\mathcal{I}$ to $\mathbb{R}$. For any positive integer $t$, we say that a set $\{I_1, \ldots, I_t\} \subseteq \mathcal{I}$ is pseudo-shattered by $\mathcal{F}$ if there exist real numbers $s_1, \ldots, s_t$ such that
$$2^t = |\{(\mathrm{sgn}(f(I_1) - s_1), \ldots, \mathrm{sgn}(f(I_t) - s_t)) : f \in \mathcal{F}\}|.$$
The *pseudo-dimension of* $\mathcal{F}$, denoted as $\mathrm{Pdim}(\mathcal{F}) \in \mathbb{N} \cup \{+\infty\}$, is the size of the largest set that can be pseudo-shattered by $\mathcal{F}$.

The main goal of statistical learning theory is to solve a problem of the following form, given a fixed parameterized function class defined by some $h$ with output space $\mathcal{O} = \mathbb{R}$:
$$\min_{\mathbf{p} \in \mathcal{P}} \; \mathbb{E}_{I \sim \mathcal{D}}[h(I, \mathbf{p})], \tag{1}$$
for an unknown distribution $\mathcal{D}$, given access to i.i.d. samples $I_1, \ldots, I_t$ from $\mathcal{D}$. In other words, one tries to "learn" the best decision $\mathbf{p}$ for minimizing an expected "score" with respect to an unknown distribution given only samples from the distribution. Such a "best" decision can be thought of as a property of the unknown distribution and the problem is to "learn" this property of the unknown distribution, only given access to samples.

The following is a fundamental result in empirical processes theory and is foundational for the above learning problem; see, e.g., Chapters 17, 18 and 19 in Anthony et al. [1999], especially Theorem 19.2.

**Theorem 2.3.** There exists a universal constant $C$ such that the following holds. Let $\mathcal{H}$ be a hypothesis class defined by some $h : \mathcal{I} \times \mathcal{P} \to \mathbb{R}$ such that the range of $h$ is in $[0, B]$ for some $B > 0$. For any distribution $\mathcal{D}$ on $\mathcal{X}$, $\epsilon > 0$, $\delta \in (0, 1)$, and
$$t \geq \frac{CB^2}{\epsilon^2} \left( \mathrm{Pdim}(\mathcal{H}) \ln \left( \frac{B}{\epsilon} \right) + \ln \left( \frac{1}{\delta} \right) \right),$$
we have
$$\left| \frac{1}{t} \sum_{i=1}^t h(I_i, \mathbf{p}) - \mathbb{E}_{I \sim \mathcal{D}} [h(I, \mathbf{p})] \right| \leq \epsilon \text{ for all } \mathbf{p} \in \mathcal{P},$$
with probability $1 - \delta$ over i.i.d. samples $I_1, \ldots, I_t \in \mathcal{I}$ of size $t$ drawn from $\mathcal{D}$.

Thus, if one solves the *sample average* problem $\min_{\mathbf{p} \in \mathcal{P}} \frac{1}{t} \sum_{i=1}^t h(I_i, \mathbf{p})$ with a large enough sample to within $O(\epsilon)$ accuracy, the corresponding $\mathbf{p}$ would solve (1) to within $O(\epsilon)$ accuracy (with high probability over the sample). Thus, the pseudo-dimension $\mathrm{Pdim}(\mathcal{H})$ is a key parameter that can be used to bound the size of a sample that is sufficient to solve the learning problem.

### 2.1.2 Neural networks

We formalize the definition of neural networks for the purposes of stating our results. Given any function $\sigma : \mathbb{R} \to \mathbb{R}$, we will use the notation $\sigma(\mathbf{x})$ for $\mathbf{x} \in \mathbb{R}^d$ to mean $[\sigma(\mathbf{x}_1), \sigma(\mathbf{x}_2), \ldots, \sigma(\mathbf{x}_d)]^{\mathsf{T}} \in \mathbb{R}^d$.

**Definition 2.4** (Neural networks). Let $\sigma : \mathbb{R} \to \mathbb{R}$ and let $L$ be a positive integer. A *neural network* with *activation* $\sigma$ and *architecture* $\boldsymbol{w} = [w_0, w_1, \ldots, w_L, w_{L+1}]^{\mathsf{T}} \in \mathbb{Z}_+^{L+2}$ is a paremterized function class, parameterized by $L + 1$ affine transformations $\{T_i : \mathbb{R}^{w_{i-1}} \to \mathbb{R}^{w_i}, i \in [L+1]\}$ with $T_{L+1}$ linear, is defined as the function

$$T_{L+1} \circ \sigma \circ T_L \circ \cdots T_2 \circ \sigma \circ T_1.$$

$L$ denotes the number of hidden layers in the network, while $w_i$ signifies the width of the $i$-th hidden layer for $i \in [L]$. The input and output dimensions of the neural network are denoted by $w_0$ and $w_{L+1}$, respectively. If $T_i$ is represented by the matrix $A^i \in \mathbb{R}^{w_i \times w_{i-1}}$ and vector $\mathbf{b}^i \in \mathbb{R}^{w_i}$, i.e., $T_i(\mathbf{x}) = A^i \mathbf{x} + \mathbf{b}^i$ for $i \in [L+1]$, then the *weights of neuron* $j \in [w_i]$ in the $i$-th hidden layer come from the entries of the $j$-th row of $A^i$ while the *bias* of the neuron is indicated by the $j$-th coordinate of $\mathbf{b}^i$. The *size* of the neural network is defined as $w_1 + \cdots + w_L$, denoted by $U$.

In the terminology of Definition 2.1, we define the neural network parameterized functions $N^\sigma : \mathbb{R}^{w_0} \times \mathbb{R}^W \to \mathbb{R}^{w_{L+1}}$, with $\mathbb{R}^{w_0}$ denoting the input space and $\mathbb{R}^W$ representing the parameter space. This parameter space is structured through the concatenation of all entries from the matrices $A^i$ and vectors $\mathbf{b}^i$, for $i \in [L+1]$, into a single vector of length $W$. The functions are defined as $N^\sigma(\mathbf{x}, \mathbf{w}) = T_{L+1}(\sigma(T_L(\cdots T_2(\sigma(T_1(\mathbf{x}))) \cdots)))$ for any $\mathbf{x} \in \mathbb{R}^{w_0}$ and $\mathbf{w} \in \mathbb{R}^W$, where each $T_i$ represents the affine transformations associated with $\mathbf{w} \in \mathbb{R}^W$.

In the context of this paper, we will focus on the following activation functions:

- **sgn:** The *Linear Threshold* (LT) activation function $\mathrm{sgn} : \mathbb{R} \to \{0, 1\}$, which is defined as $\mathrm{sgn}(x) = 0$ if $x < 0$ and $\mathrm{sgn}(x) = 1$ otherwise.

- **ReLU:** The *Rectified Linear Unit* (ReLU) activation function $\mathrm{ReLU} : \mathbb{R} \to \mathbb{R}_{\geq 0}$ is defined as $\mathrm{ReLU}(x) = \max\{0, x\}$.

- **CReLU:** The *Clipped Rectified Linear Unit* (CReLU) activation function $\mathrm{CReLU} : \mathbb{R} \to [0, 1]$ is defined as $\mathrm{CReLU}(x) = \min\{\max\{0, x\}, 1\}$.

- **Sigmoid:** The *Sigmoid* activation function $\mathrm{Sigmoid} : \mathbb{R} \to (0, 1)$ is defined as $\mathrm{Sigmoid}(x) = \frac{1}{1+e^{-x}}$.

### 2.2 Our results

In this study, we extend the framework introduced by Balcan et al. Balcan et al. [2021a] to explore the learnability of tunable algorithmic parameters through neural networks. Consider a computational problem given by a family of instances $\mathcal{I}$. Let us say we have a suite of algorithms for this problem, parameterized by parameters in $\mathcal{P}$. We also have a *score function* that evaluates how well a particular algorithm, given by specific settings of the parameters, performs on a particular instance. In other words, the score function is given by $S : \mathcal{I} \times \mathcal{P} \to [0, B]$, where $B \in \mathbb{R}_+$ determines a priori upper bound on the score. The main goal of data-driven algorithm design is to find a particular algorithm in our parameterized family of algorithms – equivalently, find a parameter setting $\mathbf{p} \in \mathcal{P}$ – that minimizes the expected score on the family of instances with respect to an unknown distribution on $\mathcal{I}$, given access to a sample of i.i.d instances from the distribution. This then becomes a special case of the general learning problem (1), where $h = S$ and one can provide precise sample complexity bounds via Theorem 2.3, if one can bound the pseudo-dimension of the corresponding hypothesis class. A bound on this pseudo-dimension is precisely the central result in Balcan et al. [2021a]; see the discussion in Section 1.

We assume the parameter space $\mathcal{P}$ is a Cartesian product of intervals $[\eta_1, \tau_1] \times \cdots \times [\eta_\ell, \tau_\ell]$, where $\eta_i \leq \tau_i$ for each $i \in [\ell]$. The transformation from the instance space $\mathcal{I}$ to the parameter space $\mathcal{P}$ is structured through the following mappings:

1. An encoder function $\mathrm{Enc} : \mathcal{I} \to \mathbb{R}^d$ is defined to convert an instance $I \in \mathcal{I}$ into a vector $\mathbf{x} = \mathrm{Enc}(I) \in \mathbb{R}^d$, facilitating the instances to be suitably processed by a neural network.

A simple example of such an encoder could be a compilation of all the instance's numerical data into a single vector; but one can allow encodings that use some predetermined features of the instances.

2. A family of neural network mappings, denoted as $N^\sigma : \mathbb{R}^d \times \mathbb{R}^W \to \mathbb{R}^\ell$, is utilized. These mappings are characterized by an activation function $\sigma$, and a fixed architecture represented by $\boldsymbol{w} = [d, w_1, \ldots, w_L, \ell] \in \mathbb{Z}_+^{L+2}$. For any given neural network parameters $\mathbf{w} \in \mathbb{R}^W$, this network maps an encoded instance $\mathbf{x} \in \mathbb{R}^d$ into $\mathbf{y} := N^\sigma(\mathbf{x}, \mathbf{w}) \in \mathbb{R}^\ell$.

3. A *squeezing activation function*, $\sigma' : \mathbb{R} \to [0, 1]$, is introduced to adjust the neural network's output to the parameter space $\mathcal{P}$. The parameter $\mathbf{p} \in \mathcal{P}$ is computed by $\mathbf{p}_i = \eta_i + (\tau_i - \eta_i)\sigma'(\mathbf{y}_i)$ for $i = 1, \ldots, \ell$.

The composite mapping from the instance space $\mathcal{I}$ to the parameter space $\mathcal{P}$ is denoted by $\varphi_{\mathbf{w}}^{N^\sigma, \sigma'}$, since the results of this study are applicable for any fixed and predetermined encoder function $\mathrm{Enc}$.

The problem of learning the best neural mapping then becomes the learning problem (1) with $h : \mathcal{I} \times \mathbb{R}^W \to \mathbb{R}$ defined by $h(I, \mathbf{w}) := S\left(I, \varphi_{\mathbf{w}}^{N^\sigma, \sigma'}(I)\right)$. We use

$$\mathcal{F}_{N^\sigma, \sigma'}^S := \left\{ S\left(\cdot, \varphi_{\mathbf{w}}^{N^\sigma, \sigma'}(\cdot)\right) : \mathcal{I} \to [0, B] \mid \mathbf{w} \in \mathbb{R}^W \right\}$$

to denote the corresponding hypothesis class (Definition 2.1).

Our first result employs linear threshold neural networks for generating algorithm parameters, inspired by their analytically tractable structure and rich expressive capabilities, as supported by findings in Khalife et al. [2023].

**Theorem 2.5.** Consider a set $\mathcal{I}$ of instances of a computational problem with a suite of algorithms parameterized by $\mathcal{P} = [\eta_1, \tau_1] \times \cdots \times [\eta_\ell, \tau_\ell]$, with score function $S : \mathcal{I} \times \mathcal{P} \to [0, B]$. Suppose that, for any given instance $I \in \mathcal{I}$, there exist at most $\Gamma$ polynomials on $\mathbb{R}^\ell$, each of degree at most $\gamma$, such that within each region of $\mathcal{P}$ where these polynomials have the same sign pattern, the function $S(I, \cdot)$ is a polynomial on $\mathbb{R}^\ell$ with degree at most $\lambda$. For linear threshold neural networks $N^{\mathrm{sgn}} : \mathbb{R}^d \times \mathbb{R}^W \to \mathbb{R}^\ell$ with a fixed architecture $\boldsymbol{w} = [d, w_1, \ldots, w_L, \ell] \in \mathbb{Z}_+^{L+2}$, having size $U$ and $W$ parameters (Definition 2.4), and using a Sigmoid squeezing function, we have

$$\mathrm{Pdim}\left(\mathcal{F}_{N^{\mathrm{sgn}}, \mathrm{Sigmoid}}^S\right) = \mathcal{O}\left(W \log(U\gamma\Gamma(\lambda + 1))\right).$$

In addition to this, we investigate the sample complexity associated with the use of $\mathrm{ReLU}$ neural networks for parameter selection.

**Theorem 2.6.** Under the same conditions as Theorem 2.5, with ReLU neural networks $N^{\mathrm{ReLU}} : \mathbb{R}^d \times \mathbb{R}^W \to \mathbb{R}^\ell$ having the same fixed architecture and clipped ReLU squeezing function, we have

$$\mathrm{Pdim}\left(\mathcal{F}_{N^{\mathrm{ReLU}}, \mathrm{CReLU}}^S\right) = \mathcal{O}\left(LW \log(U + \ell) + W \log(\gamma\Gamma(\lambda + 1))\right).$$

**Remark 2.7.** Theorem 2.6 can be easily extended to the case where general piecewise polynomial activation functions are used instead of ReLU, with each function having at most $p \geq 1$ pieces and degree at most $q \geq 1$. By applying the same proof techniques as in the theorem above and using Theorem 7 in Bartlett et al. [2019], the pseudo-dimension changes to $\mathcal{O}\left(LW \log(p(U + \ell)) + L^2 W \log(q) + W \log(\gamma\Gamma(\lambda + 1))\right)$. While we present the ReLU case as the main theorem due to its common use in practical applications, from a theoretical perspective, ReLU activation functions are not fundamentally different from other piecewise polynomial activation functions in the context of this paper.

It is not hard to adapt the proofs of Theorem 2.5 and Theorem 2.6 to show that if any dimension of the parameter space is all of $\mathbb{R}$ rather than a bounded interval, the pseudo-dimension bounds will only be smaller, under the same conditions. Additionally, if $\mathcal{P} = \{\mathbf{p}_1, \ldots, \mathbf{p}_r\}$ is a finite set, the problem can be viewed as a multi-classification problem. That is, consider a neural network $N^\sigma : \mathbb{R}^d \times \mathbb{R}^W \to \mathbb{R}^r$, where for any $\mathbf{x} \in \mathbb{R}^d$ and $\mathbf{w} \in \mathbb{R}^W$, $N^\sigma(\mathbf{x}, \mathbf{w})$ outputs an $r$-dimensional vector, and we select the parameter corresponding to the largest dimension. The pseudo-dimension of this problem is given by the following:

**Corollary 2.8.** Under the same conditions as Theorem 2.5 and 2.6, but with $\mathcal{P} = \{\mathbf{p}_1, \ldots, \mathbf{p}_r\}$,

$$\mathrm{Pdim}(\mathcal{F}_{N^{\mathrm{sgn}}, \mathrm{Sigmoid}}^S) = \mathcal{O}\left(W \log(Ur)\right) \text{ and } \mathrm{Pdim}(\mathcal{F}_{N^{\mathrm{ReLU}}, \mathrm{CReLU}}^S) = \mathcal{O}\left(LW \log(U + r)\right).$$

# 3 Application to branch-and-cut

## 3.1 Preliminaries

**Definition 3.1** (Integer linear programming (ILP)). Let $m, n \in \mathbb{N}_+$ be fixed natural numbers, and let $A \in \mathbb{Q}^{m \times n}$, $\mathbf{b} \in \mathbb{Q}^m$, $\mathbf{c} \in \mathbb{R}^n$. The integer linear programming problem is formulated as

$$\max\{\mathbf{c}^\mathsf{T}\mathbf{x} : A\mathbf{x} \leq \mathbf{b}, \mathbf{x} \geq 0, \mathbf{x} \in \mathbb{Z}^n\}.$$

The most successful algorithms and solvers for integer programming problems are based on a methodology called *branch-and-cut*. In a branch-and-cut algorithm, one maintains two things in every iteration of the algorithm: 1) a current guess for the optimal solution, 2) a collection of polyhedra that are subsets of the original polyhedral relaxation of the ILP. In every iteration, one of these polyhedra are selected and the *continuous* linear programming (LP) solution for that selected polyhedron is computed. If the solution has objective value worse than the current guess, this polyhedron is discarded from the list and the algorithm moves to the next iteration. Otherwise, if the solution is integral, the guess is updated with this integral solution and this polyhedron is removed from further consideration. If the LP solution is not integral, one decides to either add some *cutting planes* or *branch*. In the former case, additional linear constraints are added to this polyhedron under consideration without eliminating any feasible solutions. In the latter case, one selects a fractional variable $\mathbf{x}_i$ in the LP solution and partitions the current polyhedron into two polyhedra by adding constraints $\mathbf{x}_i \leq \lfloor f_i \rfloor$ and $\mathbf{x}_i \geq \lfloor f_i \rfloor + 1$, where $f_i$ is the value of this fractional variable. The current polyhedron is then replaced in the list by these two new polyhedra. This entire process can be tracked by a *branch-and-cut tree* whose nodes are precisely the different polyhedra processed by the algorithm. The algorithm terminates when there are no more polyhedra left in the active list and the current guess is reported as the optimal solution. As is often done in practice, an *a priori* bound $B$ is set on the size of a tree; if this bound is exceeded by the algorithm at any stage, the algorithm exist early and the current guess for the solution is returned. The branch-and-cut tree size is a very good indication of how long the algorithm takes to solve the problem since the main time is spent on solving the individual LPs in the iterations of the algorithm. We will thus use the tree size as the "score" function to decide how well branch-and-cut did on any instance.

There are many different strategies to generate cutting planes in branch-and-cut Conforti et al. [2014], Nemhauser and Wolsey [1988], Schrijver [1986]. We will focus on the so-called Chvátal-Gomory (CG) cutting planes and Gomory Mixed-Integer (GMI) cuts Conforti et al. [2014]. There are usually several choices of such cutting planes to add (and some families are even infinite in size Conforti et al. [2014]). We wish to apply the results of Section 2 to decide which cutting plane to select so that the branch-and-cut tree size is small.

## 3.2 Learnability of parameterized CG cut(s)

Let $m, n$ be positive integers. We consider the ILP instance space $\mathcal{I} \subseteq \{(A, \mathbf{b}, \mathbf{c}) : A \in \mathbb{Q}^{m \times n}, \mathbf{b} \in \mathbb{Q}^m, \mathbf{c} \in \mathbb{R}^n\}$, along with a fixed encoder function $\mathrm{Enc} : \mathcal{I} \to \mathbb{R}^d$. A simple encoder might stack all elements of $(A, \mathbf{b}, \mathbf{c}) \in \mathcal{I}$ into a single vector of length $d = mn + m + n$. We also impose the conditions that $\sum_{i=1}^m \sum_{j=1}^n |A_{ij}| \leq a$ and $\sum_{i=1}^m |\mathbf{b}_i| \leq b$ for any $(A, \mathbf{b}, \mathbf{c}) \in \mathcal{I}$.

Following the discussion in Balcan et al. [2021d], we define $f_{\mathrm{CG}}(I, \mathbf{u})$ as the size of the branch-and-bound tree for a given ILP instance $I \in \mathcal{I}$ with a CG cut parameterized by a multiplier $\mathbf{u} \in [0, 1]^m$ added at the root. We interpret $f_{\mathrm{CG}}$ as a score function elaborated in Section 2.2. The piecewise structure of $f_{\mathrm{CG}}$ in its parameters is characterized by:

**Lemma 3.2** (Lemma 3.2 in Balcan et al. [2021d]). For any ILP instance $I \in \mathcal{I}$, there are at most $M := 2(a+b+n)$ hyperplanes partitioning the parameter space $[0, 1]^m$ into regions where $f_{\mathrm{CG}}(I, \mathbf{u})$ remains constant for all $\mathbf{u}$ within each region.

Applying Theorem 2.5 and Theorem 2.6 to $f_{\mathrm{CG}}$ yields these pseudo-dimension bounds:

**Proposition 3.3.** Under the same conditions as Theorem 2.5 and 2.6, with the score function $f_{\mathrm{CG}}$,

$$\mathrm{Pdim}\left(\mathcal{F}_{N^{\mathrm{sgn}}, \mathrm{Sigmoid}}^{f_{\mathrm{CG}}}\right) = \mathcal{O}\left(W \log(UM)\right),$$

$$\mathrm{Pdim}\left(\mathcal{F}_{N^{\mathrm{ReLU}}, \mathrm{CReLU}}^{f_{\mathrm{CG}}}\right) = \mathcal{O}\left(LW \log(U + m) + W \log M\right).$$

Extending this to adding $k$ CG cuts sequentially, we define $f_{\text{CG}}^k(I, (\mathbf{u}_1, \ldots, \mathbf{u}_k))$ as the branch-and-bound tree size after adding a sequence of $k$ CG cuts parameterized by $\mathbf{u}_1, \ldots, \mathbf{u}_k$ at the root for a given ILP instance $I \in \mathcal{I}$. The piecewise structure of $f_{\text{CG}}^k$ in its parameters is given by:

**Lemma 3.4** (Lemma 3.4 in Balcan et al. [2021d]). *For any ILP instance $I \in \mathcal{I}$, there are $\mathcal{O}(k2^k M)$ multivariate polynomials with $mk + k(k-1)/2$ variables and degree at most $k$ partitioning the parameter space $[0,1]^{mk+k(k-1)/2}$ into regions where $f_{\text{CG}}^k(I, (\mathbf{u}_1, \ldots, \mathbf{u}_k))$ remains constant for all $(\mathbf{u}_1, \ldots, \mathbf{u}_k)$ within each region.*

Accordingly, the pseudo-dimension bounds are

**Proposition 3.5.** *Under the same conditions as Theorem 2.5 and 2.6, with the score function $f_{\text{CG}}^k$,*

$$\text{Pdim}\left(\mathcal{F}_{N^{\text{sgn}}, \text{Sigmoid}}^{f_{\text{CG}}^k}\right) = \mathcal{O}\left(W \log(UM) + Wk\right),$$

$$\text{Pdim}\left(\mathcal{F}_{N^{\text{ReLU}}, \text{CReLU}}^{f_{\text{CG}}^k}\right) = \mathcal{O}\left(LW \log(U + mk) + W \log M + Wk\right).$$

### 3.3 Learnability of cutting plane(s) from a finite set

The selection of an optimal cut from an infinite pool of candidate cuts, as discussed in Proposition 3.3 and Proposition 3.5, is often difficult and inefficient in practice. Consequently, a popular way is to select cuts based on information from the simplex tableau (such as GMI cuts), as well as some combinatorial cuts, which inherently limit the number of candidate cuts considered to be finite.

Suppose we have a finite set of cuts $\mathcal{C}$, and we define $f_{\text{ROW}}(I, c)$ as the branch-and-bound tree size after adding a cut $c \in \mathcal{C}$ at the root for a given ILP instance $I \in \mathcal{I}$. Then Corollary 2.8 implies

**Proposition 3.6.** *Under the same conditions as Theorem 2.5 and 2.6, with the score function $f_{\text{ROW}}$,*

$$\text{Pdim}\left(\mathcal{F}_{N^{\text{sgn}}, \text{Sigmoid}}^{f_{\text{ROW}}}\right) = \mathcal{O}\left(W \log(U|\mathcal{C}|)\right) \text{ and } \text{Pdim}\left(\mathcal{F}_{N^{\text{ReLU}}, \text{CReLU}}^{f_{\text{ROW}}}\right) = \mathcal{O}\left(LW \log(U + |\mathcal{C}|)\right).$$

### 3.4 Learnability of cut selection policy

One of the leading open-source solvers, SCIP Gamrath et al. [2020], uses cut selection methodologies that rely on combining several auxiliary score functions. For a finite set of cutting planes $\mathcal{C}$, instead of directly using a neural network for selection, this model selects $c^* \in \arg\max_{c \in \mathcal{C}} \sum_{i=1}^{\ell} \mu_i \text{Score}_i(c, I)$ for each instance $I \in \mathcal{I}$. Here, $\text{Score}_i$ represents different heuristic scoring functions that assess various aspects of a cut, such as "Efficacy" Balas et al. [1996] and "Parallelism" Achterberg [2007], for a specific instance $I$, and the coefficients $\boldsymbol{\mu} \in [0,1]^{\ell}$ are tunable weights for these scoring models. Since $\mathcal{C}$ is considered to be finite, the above optimization problem is solved through enumeration. The authors in Turner et al. [2023] have experimentally implemented the idea of using a neural network to map instances to weights. We provide an upper bound on the pseudo-dimension in the following proposition of this learning problem.

**Proposition 3.7.** *Under the same conditions as Theorem 2.5 and 2.6, let $f_S(I, \boldsymbol{\mu})$ denote the size of the branch-and-bound tree for $I$ after adding a cutting plane determined by the weighted scoring model parameterized by $\boldsymbol{\mu} \in [0,1]^{\ell}$. The pseudo-dimension bounds are given by:*

$$\text{Pdim}\left(\mathcal{F}_{N^{\text{sgn}}, \text{Sigmoid}}^{f_S}\right) = \mathcal{O}\left(W \log(U|\mathcal{C}|)\right),$$

$$\text{Pdim}\left(\mathcal{F}_{N^{\text{ReLU}}, \text{CReLU}}^{f_S}\right) = \mathcal{O}\left(LW \log(U + \ell) + W \log(|\mathcal{C}|)\right).$$

## 4 Discussions and open questions

In our study, we concentrated on adding CG cuts solely at the root of the branch-and-cut tree. However, devising a strategy that generates high quality cutting planes while being efficient across the entire branch-and-cut tree poses a significant and intriguing challenge for future research. Further, our theoretical findings are applicable to any encoder that maps instances into Euclidean spaces. Hence, utilizing a fixed encoder capable of converting ILPs with different number of constraints into a same Euclidean space can in principle enable the training of a unified neural network to generate cutting planes across the branch-and-cut tree. Moreover, an effective encoder could improve the

neural network's performance beyond the one achieved with the basic stacking encoder used in our paper.

The neural network training problem (2) also requires further study. We used an RL approach (see Appendix A) to update the neural network parameters and minimize the average tree size for ILP instances sampled from a given distribution. However, this method does not guarantee convergence to optimal parameters, and relies heavily on the exploratory nature of the RL algorithm. For ILP distributions where random exploration of Chvátal multipliers is unlikely to yield smaller tree sizes, the RL algorithm may struggle to identify an effective parameter setting. Developing a more efficient and robust training methodology would greatly improve the practical value of our work.

## Acknowledgments and Disclosure of Funding

All four authors gratefully acknowledge support from Air Force Office of Scientific Research (AFOSR) grant FA95502010341. Hongyu Cheng, Barara Fiedorowicz and Amitabh Basu also gratefully acknowledge support from National Science Foundation (NSF) grant CCF2006587. Hongyu Cheng also acknowledges support from the Johns Hopkins University (JHU) Mathematical Institute for Data Science (MINDS) Fellowship, the Duncan Award 24-33, and the Rufus P. Isaacs Graduate Fellowship. Barbara Fiedorowicz also acknowledges support from the JHU Duncan Award 24-31.

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

# A  Numerical experiments

In this section, for a given ILP instance space $\mathcal{I}$ conforming to the description in Section 3.2, and a fixed distribution $\mathcal{D}$ over it, we primarily attempt to employ ReLU neural networks for choosing a CG cut multiplier for each instance in the distribution, which translates into addressing the following neural network training (empirical risk minimization) problem:

$$\min_{\mathbf{w} \in \mathbb{R}^W} \frac{1}{t} \sum_{i=1}^{t} f_{\mathrm{CG}}\left(I_i, \varphi_{\mathbf{w}}^{N^{\mathrm{ReLU}}, \mathrm{CReLU}}(I_i)\right), \tag{2}$$

where $I_1, \ldots, I_t \in \mathcal{I}$ are i.i.d. samples drawn from $\mathcal{D}$, and recall that $\varphi_{\mathbf{w}}^{N^{\mathrm{ReLU}}, \mathrm{CReLU}}(\cdot) = \mathrm{CReLU}(N^\sigma(\mathrm{Enc}(\cdot), \mathbf{w}))$.

This problem, concerning the selection from an infinite pool of cuts, presents a significant challenge. The target function $f_{\mathrm{CG}}$, in relation to its parameters, is an intricately complex, high-dimensional, and piecewise constant function with numerous pieces (recall Lemma 3.2), making direct application of classical gradient-based methods seemingly impractical. As such, we use an RL approach, with our goal switched to identify a relatively "good" neural network for this distribution. Then, based on Proposition 3.3 and Theorem 2.3 the average performance of the chosen parameter setting on sampled instances closely approximates the expected performance on the distribution in high probability, given that the sample size $t$ is sufficiently large.

## A.1  Experimental setup

**Data.**  We consider the multiple knapsack problems Kellerer et al. [2004] with 16 items and 2 knapsacks, using the distribution utilized in Balcan et al. [2021b] that the authors refer to as the "Chvátal distribution" as it is inspired by Chvátal [1980]. Our synthetic dataset has a training set of 5,000 instances and a test set of 1,000 instances from the same distribution.

**Training.**  Each instance $I$ sampled from the distribution $\mathcal{D}$ is treated as a state in RL, with the outputs of the neural network considered as actions in the RL framework. The neural network thus functions as the actor in an actor-critic scheme Silver et al. [2014], where the reward is defined as the percentage reduction in the tree size after adding a cut, i.e., $\frac{f_{\mathrm{CG}}(I, \mathbf{0}) - f_{\mathrm{CG}}(I, \mathbf{u})}{f_{\mathrm{CG}}(I, \mathbf{0})}$. The Twin Delayed Deep Deterministic Policy Gradient (TD3) algorithm Fujimoto et al. [2018] is used here for the training of the neural network.

The experiments were conducted on a Linux machine with a 12-core Intel i7-12700F CPU, 32GB of RAM, and an NVIDIA RTX 3070 GPU with 8GB of VRAM. We used Gurobi 11.0.1 Gurobi Optimization, LLC [2023] to solve the ILPs, with default cuts, heuristics, and presolve settings turned off. The neural networks were implemented using PyTorch 2.3.0. The details of the implementation are available at `https://github.com/Hongyu-Cheng/LearnCGusingNN`.

## A.2  Empirical results

**Better cuts.**  The experimental results indicate that even a suboptimal neural network parameterization can outperform the cut selection methodologies used in leading solvers like SCIP. Figure 1 presents the average tree size comparison across 1,000 novel test instances, with three distinct strategies:

1. The blue solid line represents the tree size when the cut is selected based on the highest convex combination score of cut efficacy and parallelism, adjusted by a parameter $\mu \in [0, 1]$ (incremented in steps of 0.01). The candidate set of cuts includes all CG and GMI cuts generated from the appropriate rows of the optimal simplex tableau.

2. The green dash-dotted line demonstrates a notable reduction in tree size when using cuts generated through our RL approach;

3. The purple dash-dotted line follows the same approach as 2., but uses linear threshold neural networks to generate CG cut parameters. The training process uses the idea of Straight-Through Estimator (STE) Bengio et al. [2013], Yin et al. [2019].

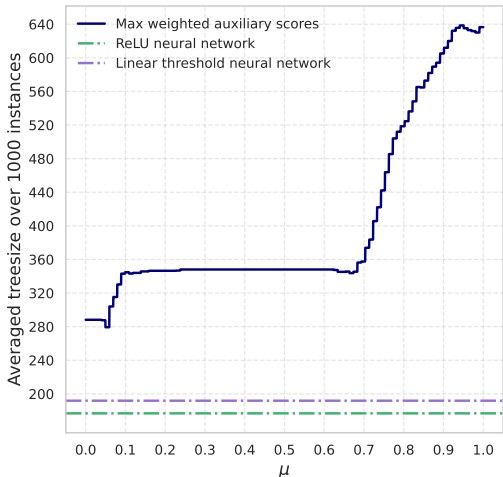

Figure 1: Comparison of branch-and-bound tree sizes using different cut selection strategies.

**Faster selection of cuts.** The cut selection using neural networks only requires matrix multiplications and ReLU activations, and is notably rapid and lends itself to efficient parallelization on GPUs. In contrast, the procedure for selecting CG and GMI cuts from a simplex tableau based on a weighted combination of cut efficacy and parallelism scores requires solving LPs to get the simplex tableaux, which is considerably slower, even without taking into account the time to score and compare potential cuts. Our empirical studies, conducted on a test set of 1,000 instances repeated 100 times, highlight the significant disparity in computational speed, as shown in Table 1.

Table 1: Comparison of the computational speeds between ReLU neural network inference and LP solving. This table presents the total time in seconds for 100 runs on a test set of 1,000 instances.

| TASKS | TIME |
|---|---|
| Computing cuts via trained neural network on GPU | 0.010 |
| Computing cuts via trained neural network on CPU | 0.055 |
| Solving required LPs using Gurobi | 2.031 |

## B  Auxiliary lemmas

**Lemma B.1.** For any $x_1, \ldots, x_n, \lambda_1, \ldots, \lambda_n > 0$, the following inequalities hold:

$$\log x_1 \leq \frac{x_1}{\lambda_1} + \log\left(\frac{\lambda_1}{e}\right), \tag{3}$$

$$x_1^{\lambda_1} \cdots x_n^{\lambda_n} \leq \left(\frac{\lambda_1 x_1 + \cdots + \lambda_n x_n}{\lambda_1 + \cdots + \lambda_n}\right)^{\lambda_1 + \cdots + \lambda_n}. \tag{4}$$

**Lemma B.2** (Theorem 5.5 in Matousek [1999], Lemma 17 in Bartlett et al. [2019], Lemma 3.3 in Anthony et al. [1999], Theorem 1.3 in Edelsbrunner [1987]). Let $\mathcal{P} \subseteq \mathbb{R}^\ell$ and let $f_1, \ldots, f_t : \mathbb{R}^\ell \to \mathbb{R}$ with $t \geq \ell$ be functions that are polynomials of degree $m$ when restricted to $\mathcal{P}$. Then

$$|\{(\text{sgn}(f_1(\mathbf{p})), \ldots, \text{sgn}(f_t(\mathbf{p}))) : \mathbf{p} \in \mathcal{P}\}| = 1, \qquad m = 0,$$

$$|\{(\text{sgn}(f_1(\mathbf{p})), \ldots, \text{sgn}(f_t(\mathbf{p}))) : \mathbf{p} \in \mathcal{P}\}| \leq \left(\frac{et}{\ell+1}\right)^{\ell+1}, \qquad m = 1,$$

$$|\{(\text{sgn}(f_1(\mathbf{p})), \ldots, \text{sgn}(f_t(\mathbf{p}))) : \mathbf{p} \in \mathcal{P}\}| \leq 2\left(\frac{2etm}{\ell}\right)^\ell, \qquad m \geq 2.$$

**Lemma B.3.** Let $h : \mathcal{I} \times \mathcal{P} \to \mathbb{R}$ define a parameterized function class with $\mathcal{P} \subseteq \mathbb{R}^\ell$, and let $\mathcal{H}$ be the corresponding hypothesis class (Definition 2.1). Let $m \in \mathbb{N}$ and $R : \mathbb{N} \to \mathbb{N}$ be a function with the following property: for any $t \in \mathbb{N}$ and $I_1, \ldots, I_t \in \mathcal{I}$, there exist $R(t)$ subsets $\mathcal{P}_1, \ldots, \mathcal{P}_{R(t)}$ of $\mathcal{P}$ such that $\mathcal{P} = \cup_{i=1}^{R(t)} \mathcal{P}_i$ and, for all $i \in [R(t)]$ and $j \in [t]$, $h(I_j, \mathbf{p})$ restricted to $\mathcal{P}_i$ is a polynomial function of degree at most $m$ depending on at most $\ell' \leq \ell$ of the coordinates. In other words, the map

$$\mathbf{p} \mapsto (h(I_1, \mathbf{p}), \ldots, h(I_t, \mathbf{p}))$$

is a piecewise polynomial map from $\mathcal{P}$ to $\mathbb{R}^t$ with at most $R(t)$ pieces. Then,

$$\mathrm{Pdim}(\mathcal{H}) \leq \sup \left\{ t \geq 1 : 2^{t-1} \leq R(t) \left( \frac{2et(m+1)}{\ell'} \right)^{\ell'} \right\}$$

*Proof.* Given any $t \in \mathbb{N}$ and $(I_1, s_i), \ldots, (I_t, s_t) \in \mathcal{I} \times \mathbb{R}$, we first bound the size of

$$\{(\mathrm{sgn}(h(I_1, \mathbf{p}) - s_1), \ldots, \mathrm{sgn}(h(I_t, \mathbf{p}) - s_t)) : \mathbf{p} \in \mathcal{P}\}.$$

Within each $\mathcal{P}_i$, $h(I_j, \mathbf{p}) - s_j$ is a polynomial in $p$ for every $j = 1, \ldots, t$, given the hypothesis. Applying Lemma B.2,

$$|\{(\mathrm{sgn}(h(I_1, \mathbf{p}) - s_1), \ldots, \mathrm{sgn}(h(I_t, \mathbf{p}) - s_t)) : \mathbf{p} \in \mathcal{P}_i\}| \leq 2 \left( \frac{2et(m+1)}{\ell'} \right)^{\ell'}.$$

Summing over the different $\mathcal{P}_i$, $i \in [R(t)]$, we obtain that

$$|\{(\mathrm{sgn}(h(I_1, \mathbf{p}) - s_1), \ldots, \mathrm{sgn}(h(I_t, \mathbf{p}) - s_t)) : \mathbf{p} \in \mathcal{P}\}| \leq 2R(t) \left( \frac{2et(m+1)}{\ell'} \right)^{\ell'}.$$

Thus, $\mathrm{Pdim}(\mathcal{H})$ is bounded by the largest $t$ such that $2^t \leq 2R(t) \left( \frac{2et(m+1)}{\ell'} \right)^{\ell'}$. $\qquad\square$

**Lemma B.4.** Let $N^{\mathrm{ReLU}} : \mathbb{R}^d \times \mathbb{R}^W \to \mathbb{R}^\ell$ be a neural network function with ReLU activation and architecture $\boldsymbol{w} = [d, w_1, \ldots, w_L, \ell]$ (Definition 2.4). Then for every natural number $t > LW$, and any $\mathbf{x}^1, \ldots, \mathbf{x}^t \in \mathbb{R}^d$, there exists subsets $\mathcal{W}_1, \ldots, \mathcal{W}_Q$ of $\mathbb{R}^W$ with $Q \leq 2^L \left( \frac{2et \sum_{i=1}^L (iw_i)}{LW} \right)^{LW}$ whose union is all of $\mathbb{R}^W$, such that $N(\mathbf{x}^j, \mathbf{w})$ restricted to $\mathbf{w} \in \mathcal{W}_i$ is a polynomial function of degree at most $L+1$ for all $(i, j) \in [Q] \times [t]$.

*Proof.* Follows from Section 2 in Bartlett et al. [1998] and Section 4 in Bartlett et al. [2019]. $\qquad\square$

**Lemma B.5** (Anthony et al. [1999], Sontag et al. [1998]). Let $N^{\mathrm{sgn}} : \mathbb{R}^{w_0} \times \mathbb{R}^W \to \mathbb{R}^\ell$ be a neural network function with sgn activation and architecture $\boldsymbol{w} = [w_0, w_1, \ldots, w_L, \ell]$, where the final linear transformation $T_{L+1}$ is taken to be the identity (Definition 2.4). Let $U = w_1 + \ldots + w_L$ denote the size of the neural network. Then for every $t \in \mathbb{N}$, and any $\mathbf{x}^1, \ldots, \mathbf{x}^t \in \mathbb{R}^{w_0}$, there exists subsets $\mathcal{W}_1, \ldots, \mathcal{W}_Q$ of $\mathbb{R}^W$ with $Q \leq \left( \frac{etU}{W'} \right)^{W'}$ whose union is all of $\mathbb{R}^W$, where $W' = \sum_{i=1}^L (w_{i-1}+1)w_i$, such that $N^{\mathrm{sgn}}(\mathbf{x}^j, \mathbf{w})$ restricted to any $\mathcal{W}_i$ is constant for all $(i, j) \in [Q] \times [t]$.

*Proof.* Fix a $t \in \mathbb{N}$ and $\mathbf{x}^1, \ldots, \mathbf{x}^t \in \mathbb{R}^{w_0}$. Consider a neuron in the first hidden layer. The output of this neuron on the input $\mathbf{x}^j$ is $\mathrm{sgn}(\langle \mathbf{a}, \mathbf{x}^j \rangle + b)$, where $\mathbf{a} \in \mathbb{R}^{w_0}, b \in \mathbb{R}$ are the weights associated with this neuron. As a function of these weights, this is a linear function, i.e., a polynomial function of degree 1. Applying Lemma B.2, there are at most $\left( \frac{et}{w_0+1} \right)^{w_0+1}$ regions in the space $(\mathbf{a}, b) \in \mathbb{R}^{w_0} \times \mathbb{R}$ such that within each region, the output of this neuron is constant. Applying the reasoning for the $w_1$ neurons in the first layer, we obtain that the output of the first hidden layer (as a $0/1$ vector in $\mathbb{R}^{w_1}$) is piecewise constant as a function of the parameters of the first layer, with at most $\left( \frac{et}{w_0+1} \right)^{(w_0+1)w_1}$ pieces. For a fixed output of the first hidden layer (which is a vector in $\mathbb{R}^{w_1}$), we can apply the same reasoning and partition space of weights of the second layer into $\left( \frac{et}{w_1+1} \right)^{(w_1+1)w_2}$ regions where the output of the second hidden layer is constant. Applying this argument iteratively across the hidden

layers, and using the inequality (4) in Lemma B.1, we deduce that a decomposition exists for $\mathbb{R}^W$ with at most

$$\left(\frac{et}{w_0+1}\right)^{(w_0+1)w_1}\left(\frac{et}{w_1+1}\right)^{(w_1+1)w_2}\cdots\left(\frac{et}{w_{L-1}+1}\right)^{(w_{L-1}+1)w_L}$$

$$\leq\left(\left(\frac{et}{w_0+1}\right)^{\frac{(w_0+1)w_1}{W'}}\left(\frac{et}{(w_1+1)}\right)^{\frac{(w_1+1)w_2}{W'}}\cdots\left(\frac{et}{w_{L-1}+1}\right)^{\frac{(w_{L-1}+1)w_L}{W'}}\right)^{W'}$$

$$\leq\left(\frac{etw_1}{W'}+\cdots+\frac{etw_L}{W'}\right)^{W'}$$

$$=\left(\frac{etU}{W'}\right)^{W'}$$

regions, such that within each such region the output of the last hidden layer of the neural network is constant, as a function of the neural network parameters, for all the vectors $\mathbf{x}^1,\ldots,\mathbf{x}^t$. $\qquad\square$

## C  Proofs of main results

*Proof of Theorem 2.5.* Let $h:\mathcal{I}\times\mathbb{R}^W\to\mathbb{R}$ as

$$h(I,\mathbf{w}):=S(I,\varphi_{\mathbf{w}}^{N^{\mathrm{sgn}},\mathrm{Sigmoid}}(I)),$$

using the notation from Section 2.2. We wish apply Lemma B.3 on the parameterized function class given by $h$ with $\mathcal{P}=\mathbb{R}^W$. Accordingly, we need to find a function $R:\mathbb{N}\to\mathbb{N}$ such that for any natural number $t>W$, and any $I_1,\ldots,I_t$, the function $\mathbf{w}\mapsto(h(I_1,\mathbf{w}),\ldots,h(I_t,\mathbf{w}))$ is a piecewise polynomial function on $\mathbb{R}^W$ with at most $R(t)$ pieces.

We consider the space of the neural parameters as a Cartesian product of the space $\mathcal{W}'$ of all the parameters of the neural network, except for the last linear transformation $T_{L+1}$, and the space $\mathbb{R}^{\ell\times w_L}$ of matrices representing this final linear transformation. Thus, we identify a one-to-one correspondence between $\mathbb{R}^W$ and $\mathcal{W}'\times\mathbb{R}^{\ell\times w_L}$.

By Lemma B.5, there exist a decomposition of $\mathcal{W}'$ into at most $\left(\frac{etU}{W'}\right)^{W'}$ regions, where $W'=W-\ell w_L$ is the number of parameters determining the space $\mathcal{W}'$, such that within each region, the output of the final hidden layer of the neural network is constant (as a function of the parameters in the region) for each input $\mathrm{Enc}(I_j),\ j\in[t]$.

We fix the parameters $\mathbf{w}\in\mathcal{W}'$ to be in one of these regions and let $\mathbf{z}^j$ be the (constant) output corresponding to input $\mathrm{Enc}(I_j)$ for any parameter settings in this region. Let us consider the behaviour of $\mathrm{Sigmoid}(A^{L+1}\mathbf{z}^j)$, which is the result of a sigmoid activation applied on the final output of the neural network, as a function of the final set of parameters encoded by the matrix $A^{L+1}\in\mathbb{R}^{\ell\times w_L}$. We follow the approach used in the proof of Theorem 8.11 in Anthony et al. [1999]. For each $k\in[\ell]$,

$$(\mathrm{Sigmoid}(A^{L+1}\mathbf{z}^j))_k=\frac{1}{1+\exp(-\sum_{i=1}^{w_L}(A_{ki}^{L+1}\mathbf{z}_i^j))}=\frac{\prod_{i=1}^{w_L}(e^{-A_{ki}^{L+1}})}{\prod_{i=1}^{w_L}(e^{-A_{ki}^{L+1}})+\prod_{i=1}^{w_L}(e^{-A_{ki}^{L+1}})^{1+\mathbf{z}_i^j}}.$$

Let $\theta_{ki}=e^{-A_{ki}^{L+1}}$ for $i\in[w_L]$, we have

$$(\mathrm{Sigmoid}(A^{L+1}\mathbf{z}^j))_k=\frac{\prod_{i=1}^{w_L}\theta_{ki}}{\prod_{i=1}^{w_L}\theta_{ki}+\left(\prod_{i=1}^{w_L}\theta_{ki}^{1+\mathbf{z}_i^j}\right)}.$$

Note that the right hand side above is a ratio of polynomials in $\theta_{ki}$ with degrees at most $2w_L$.

Next, as per the hypothesis of Theorem 2.5, let $\psi_1^j,\ldots,\psi_\Gamma^j$ be the polynomials on $\mathbb{R}^\ell$, each of degree at most $\gamma$, such that the function $S(I_j,\cdot)$ is a polynomial with degree at most $\lambda$ within each of the regions where the signs of $\psi_1^j,\ldots,\psi_\Gamma^j$ are constant. Moreover, let $T:[0,1]^\ell\to\mathcal{P}$ be the affine linear map $T(\mathbf{u})_k=\eta_k+(\tau_k-\eta_k)\mathbf{u}_k$. We observe then that for all $A^{L+1}$ such that the functions

$$\psi_1^j(T(\mathrm{Sigmoid}(A^{L+1}\mathbf{z}^j))),\ldots,\psi_\Gamma^j(T(\mathrm{Sigmoid}(A^{L+1}\mathbf{z}^j)))$$

have the same signs, then $h(I_j, (\mathbf{w}, A^{L+1})) = S(I_j, \varphi_{\mathbf{w}, A^{L+1}}^{N^{\mathrm{sgn}}, \mathrm{Sigmoid}}(I_j))$ is a polynomial of degree at most $\lambda$. By the observations made above, the functions $\psi_1^j(T(\mathrm{Sigmoid}(A^{L+1}\mathbf{z}^j))), \ldots, \psi_\Gamma^j(T(\mathrm{Sigmoid}(A^{L+1}\mathbf{z}^j)))$ are rational functions, i.e., ratios of polynomials, in the transformed parameters $\theta_{ki}$ and the numerators and denominators of these rational functions have degrees bounded by $2w_L\gamma$. Since $\mathrm{sgn}\left(\frac{P}{Q}\right) = \mathrm{sgn}(PQ)$ for any multivariate polynomials $P, Q$ (whenever the denominator is nonzero), we can bound the total number of sign patterns for $\psi_1^j(T(\mathrm{Sigmoid}(A^{L+1}\mathbf{z}^j))), \ldots, \psi_\Gamma^j(T(\mathrm{Sigmoid}(A^{L+1}\mathbf{z}^j)))$ using Lemma B.2 where the polynomials defining the regions have degree at most $3w_L\gamma$ on the transformed parameters $\theta_{ki}$. We have to consider all the functions $\psi_1^j(T(\mathrm{Sigmoid}(A^{L+1}\mathbf{z}^j))), \ldots, \psi_\Gamma^j(T(\mathrm{Sigmoid}(A^{L+1}\mathbf{z}^j)))$ for $j \in [t]$, giving us a total of $t\Gamma$ rational functions. Thus, an application of Lemma B.2 gives us a decomposition of $\mathbb{R}^{\ell \times w_L}$ into at most

$$2\left(\frac{2e \cdot t\Gamma \cdot 3\gamma w_L}{\ell w_L}\right)^{\ell w_L} \leq 2\left(\frac{6et\gamma\Gamma}{\ell}\right)^{\ell w_L}$$

regions such that $h(I_j, (\mathbf{w}, A^{L+1}))$ is a polynomial of degree at most $\lambda$ within each region. Combined with the bound $\left(\frac{etU}{W'}\right)^{W'}$ on the number of regions for $\mathbf{w} \in \mathcal{W}'$, we obtain a decomposition of the full parameter space $\mathcal{W}' \times \mathbb{R}^{\ell \times w_L}$ into at most

$$R(t) := \left(\frac{etU}{W'}\right)^{W'} \cdot 2\left(\frac{6et\gamma\Gamma}{\ell}\right)^{\ell w_L}$$

regions, such that within each region $h(I_j, (\mathbf{w}, A^{L+1}))$, as a function of $(\mathbf{w}, A^{L+1})$, is a polynomial of degree at most $\lambda$, for every $j \in [t]$. Moreover, note that within each such region, $h(I_j, (\mathbf{w}, A^{L+1}))$ depends only on $A^{L+1}$. Applying Lemma B.3, $\mathrm{Pdim}\left(\mathcal{F}_{N^{\mathrm{sgn}}, \mathrm{Sigmoid}}^S\right)$ is bounded by the largest $t \in \mathbb{N}$ such that

$$2^{t-1} \leq \left(\frac{etU}{W'}\right)^{W'} \cdot 2\left(\frac{6et\gamma\Gamma}{\ell}\right)^{\ell w_L} \cdot \left(\frac{2et(\lambda+1)}{\ell w_L}\right)^{\ell w_L}.$$

Taking logarithms on both sides, we want the largest $t$ such that

$$\frac{1}{2}(t-2) \leq W'\log\left(\frac{etU}{W'}\right) + \ell w_L \log\left(\frac{6et\gamma\Gamma}{\ell}\right) + \ell w_L \log\left(\frac{2et(\lambda+1)}{\ell w_L}\right)$$

As we only need to derive an upper bound for the pseudo-dimension, we can loosen the inequality above using the inequality (3) in Lemma B.1:

$$\begin{aligned}
\frac{1}{2}(t-2) \leq & W'\left(\frac{etU/W'}{8eU} + \log(8U)\right) + \ell w_L \left(\frac{6et\gamma\Gamma/\ell}{48e\gamma\Gamma w_L} + \log(48\gamma\Gamma w_L)\right) \\
& + \ell w_L \left(\frac{2et(\lambda+1)/(\ell w_L)}{16e(\lambda+1)} + \log(16(\lambda+1))\right) \\
\leq & \frac{1}{8}t + W'\log(8U) + \frac{1}{8}t + \ell w_L \log(48\gamma\Gamma w_L) + \frac{1}{8}t + \ell w_L \log(16(\lambda+1)) \\
\leq & \frac{3}{8}t + W\log(8U) + \ell w_L \log\left(\frac{48\gamma\Gamma w_L \cdot 16(\lambda+1)}{8U}\right) \\
\leq & \frac{3}{8}t + W\log(8U) + \ell w_L \log(96\gamma\Gamma(\lambda+1)w_L/U)
\end{aligned}$$

then it's not hard to see that

$$\mathrm{Pdim}\left(\mathcal{F}_{N^{\mathrm{sgn}}, T\circ\mathrm{Sigmoid}}^S\right) = \mathcal{O}\left(W\log U + \ell w_L \log\left(\gamma\Gamma(\lambda+1)\right)\right) = \mathcal{O}\left(W\log(U\gamma\Gamma(\lambda+1))\right).$$

$\square$

*Proof of Theorem 2.6.* Let $h : \mathcal{I} \times \mathbb{R}^W \to \mathbb{R}$ as

$$h(I, \mathbf{w}) := S(I, \varphi_{\mathbf{w}}^{N^{\mathrm{ReLU}}, \mathrm{CReLU}}(I)),$$

using the notation from Section 2.2. We wish apply Lemma B.3 on the parameterized function class given by $h$ with $\mathcal{P} = \mathbb{R}^W$. Accordingly, we need to find a function $R : \mathbb{N} \to \mathbb{N}$ such that for any natural number $t > LW$, and any $I_1, \ldots, I_t$, the function $\mathbf{w} \mapsto (h(I_1, \mathbf{w}), \ldots, h(I_t, \mathbf{w}))$ is a piecewise polynomial function on $\mathbb{R}^W$ with at most $R(t)$ pieces.

Note that $\varphi_{\mathbf{w}}^{\mathrm{ReLU},\mathrm{CReLU}}(I)$ can be seen as the output of a neural network with ReLU activations and architecture $[d, w_1, \ldots, w_L, 2\ell, \ell]$, where the final linear function is the fixed function $T(\mathbf{u})_k = (\tau_k - \eta_k)\mathbf{u}_k$. This is because $\mathrm{CReLU}(x) = \mathrm{ReLU}(x) - \mathrm{ReLU}(x - 1)$ can be simulated using two ReLU neurons. Applying Lemma B.4, this implies that given $t \in \mathbb{N}$ and $I_1, \ldots, I_t \in \mathcal{I}$, there are

$$Q \leq 2^{L+1} \left( \frac{2et \cdot \sum_{i=1}^{L+1}(iw_i)}{(L+1)W} \right)^{(L+1)W} \leq 2^{L+1} \left( \frac{2et(U + 2\ell)}{W} \right)^{(L+1)W}$$

regions $\mathcal{W}_1, \ldots, \mathcal{W}_Q$ whose union is all of $\mathbb{R}^W$, such that $\varphi_{\mathbf{w}}^{N^{\mathrm{ReLU}},\mathrm{CReLU}}(I_j)$ restricted to $\mathbf{w} \in \mathcal{W}_i$ is a polynomial function of degree at most $L + 2$ for all $(i, j) \in [Q] \times [t]$.

Next, as per the hypothesis of Theorem 2.5, let $\psi_1^j, \ldots, \psi_\Gamma^j$ be the polynomials on $\mathbb{R}^\ell$, each of degree at most $\gamma$, such that the function $S(I_j, \cdot)$ is a polynomial with degree at most $\lambda$ within each of the regions where the signs of $\psi_1^j, \ldots, \psi_\Gamma^j$ are constant. Thus, for all $\mathbf{w} \in \mathbb{R}^W$ such that

$$\psi_1^j(\varphi_{\mathbf{w}}^{\mathrm{ReLU},\mathrm{CReLU}}(I_j)), \ldots, \psi_\Gamma^j(\varphi_{\mathbf{w}}^{\mathrm{ReLU},\mathrm{CReLU}}(I_j))$$

have the same signs, then $h(I_j, \mathbf{w}) = S(I_j, \varphi_{\mathbf{w}}^{\mathrm{ReLU},\mathrm{CReLU}}(I_j))$ is a polynomial of degree at most $\lambda(L + 2)$. The functions $\psi_1^j(\varphi_{\mathbf{w}}^{\mathrm{ReLU},\mathrm{CReLU}}(I_j)), \ldots, \psi_\Gamma^j(\varphi_{\mathbf{w}}^{\mathrm{ReLU},\mathrm{CReLU}}(I_j))$ are polynomials of degree at most $\gamma(L+2)$. Considering all these polynomials for $j \in [t]$, by Lemma B.2, each $\mathcal{W}_i \subseteq \mathbb{R}^W$ from the decomposition above can be further decomposed into at most $2\left( \frac{2et\Gamma\gamma(L+2)}{W} \right)^W$ regions such that for all $\mathbf{w}$ in such a region, $h(I_j, \mathbf{w})$ is a polynomial function of $\mathbf{w}$ of degree at most $\lambda(L+2)$.

To summarize the arguments above, we have a decomposition of $\mathbb{R}^W$ into at most

$$R(t) := 2^{L+1} \left( \frac{2et(U + 2\ell)}{W} \right)^{(L+1)W} \cdot 2\left( \frac{2et\Gamma\gamma(L+2)}{W} \right)^W$$

regions such that within each region, $h(I_j, \mathbf{w})$ is a polynomial function of $\mathbf{w}$ of degree at most $\lambda(L + 2)$. Applying Lemma B.3, $\mathrm{Pdim}\left( \mathcal{F}_{N^{\mathrm{ReLU}},\mathrm{CReLU}}^S \right)$ is bounded by the largest $t \in \mathbb{N}$ such that

$$2^{t-1} \leq 2^{L+1} \left( \frac{2et(U + 2\ell)}{W} \right)^{(L+1)W} \cdot 2\left( \frac{2et\Gamma\gamma(L+2)}{W} \right)^W \cdot \left( \frac{2et(\lambda+1)(L+2)}{W} \right)^W,$$

which is bounded by the largest $t$ such that

$$\begin{aligned}
\frac{1}{2}(t - 1) \leq{} & L + 2 + (L+1)W\log\left( \frac{2et(U+2\ell)}{W} \right) + W\log\left( \frac{2et\Gamma\gamma(L+2)}{W} \right) \\
& + W\log\left( \frac{2et(\lambda+1)(L+2)}{W} \right) \\
\leq{} & L + 2 + \frac{1}{8}t + (L+1)W\log(16(U+2\ell)) + \frac{1}{8}t + W\log(16\Gamma\gamma(L+2)) \\
& + \frac{1}{8}t + W\log(16(\lambda+1)(L+2)) \\
\leq{} & L + 2 + \frac{3}{8}t + (L+1)W\log(16(U+2\ell)) + W\log(\gamma\Gamma(\lambda+1)) + 2W\log(16(L+2)),
\end{aligned}$$

where the inequality (3) in Lemma B.1 is applied in the second line. Then it's not hard to see that

$$\mathrm{Pdim}\left( \mathcal{F}_{N^{\mathrm{ReLU}},\mathrm{CReLU}}^S \right) = \mathcal{O}\left( LW\log(U+\ell) + W\log(\gamma\Gamma(\lambda+1)) \right).$$

$\square$

*Proof of Corollary 2.8.* We introduce an auxiliary function $f : \mathbb{R}^r \to \{\mathbf{p}_1, \ldots, \mathbf{p}_r\}$ given by $f(\mathbf{x}) = \mathbf{p}_{\arg\max_{i \in [r]} \mathbf{x}_i}$, and let $S' : \mathcal{I} \times \mathbb{R}^r \to \mathbb{R}$ be

$$S'(I, \mathbf{x}) := S(I, f(\mathbf{x})).$$

There exists a decomposition of the $\mathbb{R}^r$ space obtained by at most $\frac{r(r-1)}{2}$ hyperplanes

$$\{\mathbf{x} \in \mathbb{R}^r : \mathbf{x}_i = \mathbf{x}_j\}, \quad \forall (i, j) \in [r] \times [r], i \neq j.$$

Within each decomposed region, the largest coordinate of $\mathbf{x}$ is unchanged. Therefore, for any fixed $I \in \mathcal{I}$, the new score function $S'(I, \cdot)$ remains constant in each of these regions. Then a direct application of Theorem 2.5 and Theorem 2.6 to $S'$ yields the desired result. $\qquad \square$

*Proof of Proposition 3.7.* The proof of Proposition 3.7 is analogous to the proof of Theorem 4.1 in Balcan et al. [2021d]. For any fixed instance $I \in \mathcal{I}$, let the set of candidate cutting planes be $\mathcal{C} = \{c_1, \ldots, c_{|\mathcal{C}|}\}$. Comparing the overall scores for all cuts introduces the following hyperplanes:

$$\sum_{i=1}^{\ell} \boldsymbol{\mu}_i \, \mathrm{Score}_i(c_j, I) = \sum_{i=1}^{\ell} \boldsymbol{\mu}_i \, \mathrm{Score}_i(c_k, I), \quad \forall j, k \in [|\mathcal{C}|], j \neq k.$$

There are at most $\frac{|\mathcal{C}|(|\mathcal{C}|-1)}{2}$ hyperplanes decomposing the $\boldsymbol{\mu}$ space $[0,1]^\ell$. Within each region defined by these hyperplanes, the selected cut remains the same, so the branch-and-cut tree size is constant. This proves that $f_S(I, \boldsymbol{\mu})$ is a piecewise constant function on $\boldsymbol{\mu}$, for any fixed $I$. We then apply Theorem 2.5 and Theorem 2.6 to derive the pseudo-dimension bounds. $\qquad \square$

