# OpenReview forum: "Sample Complexity of Algorithm Selection Using Neural Networks and Its Applications to Branch-and-Cut"
_NeurIPS.cc/2024/Conference — NeurIPS 2024 poster_

### Official Review · Reviewer_FoeV · 2024-07-08

**Soundness:** 3
**Presentation:** 2
**Contribution:** 2
**Rating:** 5
**Confidence:** 4

**Summary:**

The paper introduces an approach to data-driven algorithm design by leveraging neural networks to select the most appropriate algorithm for a given instance of a computational problem. The motivation is clear. This paper decision to extend the paradigm from selecting a single best algorithm to a context-specific selection. The introduction of neural networks as the parameterized family of mappings is a novel approach that could potentially offer significant benefits in terms of adaptability and performance. Overall, the paper presents an interesting approach to algorithm selection and parameter tuning using neural networks. The theoretical grounding and practical motivation are commendable.

**Strengths:**

The methodology of using neural networks to map problem instances to the most suitable algorithm is well-articulated. The derivation of rigorous sample complexity bounds is a strong theoretical contribution, providing a solid foundation for the practical applications discussed later. The integration of these theoretical insights with practical computational experiments is a strength of the paper.

**Weaknesses:**

(1) Training Overhead: Training neural networks can be computationally intensive, especially when dealing with large and complex datasets. The time and resources required for training may limit the scalability of the approach.

(2) Inference Time: While the method aims to optimize performance on a per-instance basis, the inference time for selecting the optimal parameters using a neural network could be longer than traditional static parameter selection methods.

**Questions:**

(1) Can the author elaborate on the role of theoretical guarantees such as the bounds on pseudo-dimension and sample complexity? How do these theoretical results translate into practical performance, and what are the limitations of the theoretical framework?

(2) For neural networks with different activation functions, this paper gives different bounds. What results will be produced when there are different activation functions in the neural network?

(3) Neural networks are often considered black boxes, making it difficult to understand why a particular parameter set was chosen for a given instance. How does the sample complexity of theoretical analysis guide the learning of specific algorithms?

**Limitations:**

While the paper mentions deriving rigorous sample complexity bounds, the tightness of these bounds and their practical implications need to be carefully evaluated. The theoretical guarantees may not always translate directly to strong empirical performance.

---

> ### Author Rebuttal · Authors · 2024-08-07
>
> Thank you very much for your thoughtful review.
>
> **Weaknesses:**
>
> 1. While training can be expensive, the neural network can then be directly used on all future instances from this distribution. There is no need to do any further analysis on future instances. Compare this to the use of (weighted sums of) auxiliary scores, as has been traditionally done in previous analysis. For every new instance, one has to find the parameter that minimizes this auxiliary score. This auxiliary optimization problem can take a much longer time than the inference time for the learned neural network. See Table 1 in our Appendix A, where we show that the inference time is much smaller than solving the corresponding LP, which is usually a prerequisite to computing and optimizing these auxiliary scores.
>
> 2. Indeed, there is a trade-off between the gain obtained by instance dependent parameters and the inference time. We believe that the inference times are very small and we get a big gain in performance (much smaller branch-and-cut trees) to compensate.
>
> **Questions:**
>
> 1. The aim of proving sample complexity bounds is to show that one can indeed learn good decisions within branch-and-cut in a principled way, and to give guarantees on the generalization error. Without such bounds, using neural networks (or any other learning method) may run into the problem that the amount of data needed to learn something useful is much larger than is practical for the problem at hand. Our bounds show that the question of learning neural mappings for integer programming is not out of reach in terms of the data needed to achieve good generalization guarantees.
>
>     A useful practical insight to draw from the sample complexity bounds is that the number of data points needed is linear in the number of neural network parameters. This means that if the practitioner wishes to use a larger neural network for even better error guarantees, then the sample size needs to increase roughly by the same factor as the increase in the number of neural network parameters.
>
> 2. It is hard to say what bounds one can achieve for other activations without doing the computations. We believe that our proof *techniques* are general enough that they can be adapted to do the calculations for other activations as well. We illustrate with piecewise polynomial activations:
>
>     Consider the same conditions as Theorem 2.6, but piecewise polynomial activation functions, each having at most $p \geq 1$ pieces and degree at most $d \geq 1$. By using the same proof techniques and applying Theorem 7 in Bartlett et al. (2019), the pseudo dimension is given by
>     $\mathcal{O}(LW \log(p(U+\ell)) + L^2W \log(d) + W \log(\gamma \Gamma(\lambda + 1))).$
>
>     We will replace Theorem 2.6 by this more general result in the final version of the paper, as well as the corresponding propositions/corollaries.
>
> 3. This is a good question. The sample complexity only gives a guarantee on the generalization error, but it provides no insight into the nature of the mapping that is learned from data. This raises a completely different, and very important and relevant, set of questions that we do not touch upon in this paper.
>
> **Limitations:**
>
> Indeed proving complementary lower bounds on the sample complexity would be good.
>
> We agree that, in general, theoretical guarantees do not always translate into practical performance. Nevertheless, our preliminary experiments show that this approach holds promise in improving modern MIP solvers, especially since the choice of parameters within branch-and-cut is still not well understood, and may be too hard a problem to be fully resolved analytically. The learning approach then provides a good way to use historical data to find good parameters. Sample complexity bounds give guarantees on the quality of the parameters learnt, which we think is valuable.

---

> > ### Comment · Reviewer_FoeV · 2024-08-12
> >
> > I appreciate the authors' thorough response. They have effectively addressed several of my concerns, leading me to raise my score to 5.

---

### Official Review · Reviewer_vvGd · 2024-07-13

**Soundness:** 3
**Presentation:** 4
**Contribution:** 3
**Rating:** 7
**Confidence:** 2

**Summary:**

This paper studies data-driven algorithm design, which learns a class of algorithms to optimize the expected performance scores given i.i.d. samples of instances.
In particular, the paper studies the sample complexity of learning (the weights of) neural networks computing the size of certain branch-and-cut trees for mixed-integer programming.

Based on the results of Balcan et al. to bound pseudo-dimension, this paper studies learning neural network weights to compute the size of certain branch-and-cut trees, avoiding the need to optimize for weighted combinations of auxiliary score functions for each new instance, and getting smaller tree sizes.

**Strengths:**

1. The paper is well written, including the context of research (to motivate data-driven algorithms and mixed-integer programming), overview, and technical aspects (neural networks, the technical definition of pseudo-dimensions).
2. The idea to study data-driven algorithm designs using neural networks to solve mixed-integer programming is new (unlike previous works studying weighted combinations of auxiliary score functions), and the result applies to a general class of neural networks.
3. The technical argument (to bound pseudo-dimension of neural networks) is intuitive, given the existing works to study data-driven algorithm design with pseudo-dimension.
4. The theoretical improvement is experimentally verified on a synthetic dataset with thousands of instances.

**Weaknesses:**

1. The application to the size of branch-and-cut tree for integer linear programming with Chvátal-Gomory cuts (parameterized by multipliers at *root*), or with Gomory Mixed-Integer cuts (at *root*), are somewhat limiting (as pointed out in the Discussion section). This reader thinks that the applications are chosen so that the geometric analysis argument can go through to bound the pseudo-dimension (hence to apply Theorems 2.5 and 2.6, Corollary 2.7), and unlikely to generalize to effective bounds in more general settings, where the geometric picture will be more complicated, which needs to account for both the neural netorks and (the nodes inside) the branch-and-cut trees.
2. The theoretical bounds are asymptotic which are unlikely to have effective/small constants.

Nitpick:
* In some places, the paper uses branch-and-*bound* tree, while in most places (including the title) the paper uses branch-and-*cut*. It may help to use a consistent name.

**Questions:**

For the original question—solving mixed-integer programming but not only to minimize branch-and-cut tree sizes—what does the results of the paper imply?

**Limitations:**

This is a theoretical paper that do not have broader societal impacts.

---

> ### Author Rebuttal · Authors · 2024-08-06
>
> Thank you very much for your thoughtful review.
>
> **Weaknesses:**
>
> - We think this opens up a whole new field of investigation at the intersection of integer programming and learning theory. We are currently working on other papers where the theory is extended to other families of cutting planes, as well as evaluating the effect at deeper nodes in the branch-and-cut tree. From our studies so far, it does not look like the analysis is easier because of the structure Chvátal-Gomory or GMI cuts. In fact, in previous work of Balcan et al (2022) already, the piecewise polynomial structure of the score function that is crucial for bounding the pseudo-dimension was shown to exist for very general cutting planes. Understanding the effect in deeper nodes of the tree is indeed trickier. We are actively exploring this very interesting problem.
>
> - These constants can be explicitly tracked. The constant for Theorem 2.3 is on the order of $1000$, and the constants of the pseudo dimension results in Sections 2 and 3 are on the order of $100$.
>
> **Questions:**
>
> We would like to clarify that the neural networks don't compute the size of the branch-and-cut trees directly, but rather output the "best" instance-dependent cutting planes to produce small branch-and-cut trees on average.
>
> Branch-and-cut tree size is generally considered a good proxy for the overall running time of the branch-and-cut algorithm. This is because most of the time is spent in solving the linear relaxations at the different nodes of the tree. Thus, minimizing the expected branch-and-cut tree size should correlate well with minimizing the overall expeected running time on the given family/distribution of instances. Having said that, there are definitely several other factors that go into the overall running time, e.g. dense versus sparse cuts etc. Your question does provide good directions to pursue in future research, by tracking other measures beyond tree size through this learning theory lens.

---

> > ### Comment · Reviewer_vvGd · 2024-08-13
> >
> > I acknowledge the rebuttal.

---

### Official Review · Reviewer_vkrW · 2024-07-14

**Soundness:** 4
**Presentation:** 3
**Contribution:** 3
**Rating:** 7
**Confidence:** 3

**Summary:**

The authors study the problem of algorithm selection for specific instances of a problem. In particular they provide theoretical results for the learnability of a mapping from problem instances to algorithm parameters with application to branch-and-cut. They establish sample complexity bounds for learning a few different neural network architectures for this task. The appendix includes some synthetic experiments where they apply this method.

**Strengths:**

The paper is theoretically rigorous and studies a significant problem (algorithm selection for problem instances). The use of a data-driven / ML approach is interesting and appears novel. It appears to this reader to make a signficant theoretical contribution.

**Weaknesses:**

- although sample complexity bounds are valuable, their practical interpretation for large scale problems is unclear. relating the established bounds to the state of practice for branch and cut, for example, might help shed some light on the importance to practitioners.

- the actual neural network training is done via reinforcement learning, suggesting sample complexity is only one of many challenges to be addressed in generating high-utility NNs to solve this mapping problem.

- the paper focuses on branch-and-cut methods for mixed-integer optimization, and generalization to other algorithmic problems is not thoroughly explored.

- the work is centrally focused on theory, so naturally the empirical contribution is minimal, essentially left to the appendix.

Miscellaneous:

- the algorithm exist early -> exits early

**Questions:**

- Theorems 2.5 and 2.6 assume that the parameter space can be decomposed into regions where the function remains polynomial with bounded degree. Is this realistic?
- Besides tree size, have you evaluated other measures of performance for branch and cut or other problems?
- What is the practical interpretation of the established bounds for algorithm selection on this problem? How does it inform network design / historical instance curation ?

**Limitations:**

The authors discuss the limitations appropriately in the conclusion.

---

> ### Author Rebuttal · Authors · 2024-08-06
>
> Thank you very much for your thoughtful review.
>
> **Weaknesses:**
>
> - A useful insight to draw from the sample complexity bounds is that the number of data samples is linear in the number of neural network parameters. This means that if the practitioner wishes to use a larger neural network for even better error guarantees, then the sample size needs to increase roughly by the same factor as the increase in the number of neural network parameters.
>
> - We agree with your assessment that sample complexity is only one aspect of understanding this complex learning problem; nevertheless, we believe sample complexity is a good first step towards obtaining a principled understanding. There is indeed a lot more to explore here, both theoretically and computationally.
>
> - Learning within branch-and-cut is a very challenging problem already with dozens of papers dedicated to it appearing in the last decade. Since branch-and-cut is both a challenging and a high impact problem, we focused our attention on this single use case of our general results to keep our message crisp. Other algorithmic problems merit separate careful investigation and should probably have dedicated papers to do full justice.
>
> - The paper is indeed focused on developing the theory. The experimental section is presented more as a proof of concept. A thorough experimental evaluation is a whole project in itself and merits a separate paper, in our opinion.
>
> **Questions:**
>
> - The polynomial structure actually has wide applicability across several problem domains. This was exploited in the STOC 2021 paper "How much data is sufficient to learn high-performing algorithms?" by Balcan et al (see our bibliography). Several important problems, ranging from integer programming to computational biology, were shown to have this structure.
>
> - We have not analyzed other measures, but most measures from the optimization literature should be amenable to our proof techniques. This will indeed form the basis for our future work.
>
> - The sample complexity bounds in this paper show how the size of the data needs to grow as we change the architecture/number of parameters in the neural network. Changing the architecture or number of parameters allows the neural network to be more or less expressive, so the overall expected score (e.g., tree sizes for branch-and-cut) will be smaller or larger, respectively, for the optimal setting of the neural parameters. However, with larger neural networks the data size has to correspondingly grow so that the ERM solution has performance close to this optimal neural network, with high probability. The analysis shows that the data size should grow linearly with the number of parameters in the neural network. Thus, such bounds allow us to quantitatively track the tradeoff between how much error (bias) one has in the neural network architecture versus how much data one needs to generalize well.

---

> > ### Comment · Reviewer_vkrW · 2024-08-12
> >
> > I thank the authors for their response. I think the insight is interesting (sample complexity bound growing linearly in the number of NN parameters) given that this isn't empirically the case in other domains (e.g. the "scaling laws" literature, which is totally experiment-based). I think the authors have made meaningful progress on this problem and i've raised my score to indicate that.

---

### Decision · Program_Chairs · 2024-09-25

**Decision:**

Accept (poster)

**Comment:**

The paper contributes to the development of the data-driven algorithm design area by providing theoretical results (sample complexity bounds ) for the learnability (via neural networks) of a mapping from problem instances to branch-and-cut algorithm parameters.
Notwithstanding issues about the real efficiency that can be obtained by exploiting the bounds, mainly linked to the training process of the neural network, the proposed results are timely and significant.
Rebuttal contributed to clarify several of the related issues raised by reviewers, leading some of them to raise their scores. Moreover, in the AC/reviewers discussion the most negative reviewer (borderline accept) was in agreement with the proposal to accept the paper.